# CaPC Learning: Confidential and Private Collaborative Learning

**Christopher A. Choquette-Choo,**[*] **Natalie Dullerud,**[*] **Adam Dziedzic**[*]
University of Toronto and Vector Institute
`{christopher.choquette.choo,natalie.dullerud}@mail.utoronto.ca`
`ady@vectorinstitute.ai`

**Yunxiang Zhang**[*][†]
The Chinese University of Hong Kong
`yunxiang.zhang@ie.cuhk.edu.hk`

**Somesh Jha**[‡]
University of Wisconsin-Madison and XaiPient
`jha@cs.wisc.edu`

**Nicolas Papernot**[‡]
University of Toronto and Vector Institute
`nicolas.papernot@utoronto.ca`

**Xiao Wang**[‡]
Northwestern University
`wangxiao@cs.northwestern.edu`

## Abstract

Machine learning benefits from large training datasets, which may not always be possible to collect by any single entity, especially when using privacy-sensitive data. In many contexts, such as healthcare and finance, separate parties may wish to collaborate and learn from each other's data but are prevented from doing so due to privacy regulations. Some regulations prevent explicit sharing of data between parties by joining datasets in a central location (*confidentiality*). Others also limit implicit sharing of data, e.g., through model predictions (*privacy*). There is currently no method that enables machine learning in such a setting, where both confidentiality and privacy need to be preserved, to prevent both explicit and implicit sharing of data. Federated learning only provides confidentiality, not privacy, since gradients shared still contain private information. Differentially private learning assumes unreasonably large datasets. Furthermore, both of these learning paradigms produce a central model whose architecture was previously agreed upon by all parties rather than enabling collaborative learning where each party learns and improves their own local model. We introduce *Confidential and Private Collaborative* (CaPC) learning, the first method provably achieving both *confidentiality* and *privacy* in a *collaborative* setting. We leverage secure multi-party computation (MPC), homomorphic encryption (HE), and other techniques in combination with privately aggregated teacher models. We demonstrate how CaPC allows participants to collaborate without having to explicitly join their training sets or train a central model. Each party is able to improve the accuracy and fairness of their model, even in settings where each party has a model that performs well on their own dataset or when datasets are *not IID* and model architectures are *heterogeneous* across parties.[1]

## 1 Introduction

The predictions of machine learning (ML) systems often reveal private information contained in their training data (Shokri et al., 2017; Carlini et al., 2019) or test inputs. Because of these limitations, legislation increasingly regulates the use of personal data (Mantelero, 2013). The relevant ethical

---

[*]Equal contributions, authors ordered alphabetically.
[†]Work done while the author was at Vector Institute.
[‡]Equal contributions, authors ordered alphabetically.
[1]Code is available at: https://github.com/cleverhans-lab/capc-iclr.

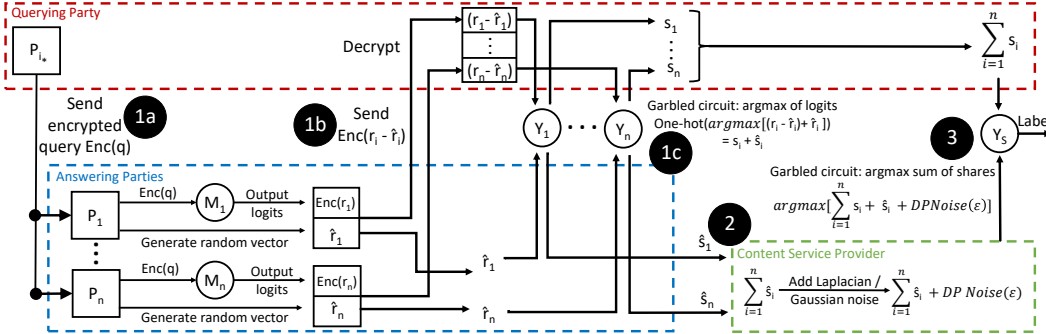

Figure 1: Confidential and Private Collaborative (CaPC) Learning Protocol: **1a** Querying party $\mathcal{P}_{i_*}$ sends encrypted query $q$ to each answering party $\mathcal{P}_i$, $i \neq i_*$. Each $\mathcal{P}_i$ engages in a secure 2-party computation protocol to evaluate $\mathsf{Enc}(q)$ on $\mathcal{M}_i$ and outputs encrypted logits $\mathsf{Enc}(\boldsymbol{r}_i)$. **1b** Each answering party, $\mathcal{P}_i$, generates a random vector $\hat{\boldsymbol{r}}_i$, and sends $\mathsf{Enc}(\boldsymbol{r}_i - \hat{\boldsymbol{r}}_i)$ to the querying party, $\mathcal{P}_{i_*}$, who decrypts to get $\boldsymbol{r}_i - \hat{\boldsymbol{r}}_i$. **1c** Each answering party $\mathcal{P}_i$ runs Yao's garbled circuit protocol ($Y_i$) with querying party $\mathcal{P}_{i_*}$ to get $\boldsymbol{s}_i$ for $\mathcal{P}_{i_*}$ and $\hat{\boldsymbol{s}}_i$ for $\mathcal{P}_i$ s.t. $\boldsymbol{s}_i + \hat{\boldsymbol{s}}_i$ is the one-hot encoding of argmax of logits. **2** Each answering party sends $\hat{\boldsymbol{s}}_i$ to the privacy guardian (PG). The PG sums $\hat{\boldsymbol{s}}_i$ from each $\mathcal{P}_i$ and adds Laplacian or Gaussian noise for DP. The querying party sums $\boldsymbol{s}_i$ from each $Y_i$ computation. **3** The PG and the querying party run Yao's garbled circuit $Y_s$ to obtain argmax of querying party and PG's noisy share. The label is output to the querying party.

concerns prompted researchers to invent ML algorithms that protect the privacy of training data and confidentiality of test inputs (Abadi et al., 2016; Konečný et al., 2016; Juvekar et al., 2018).

Yet, these algorithms require a large dataset stored either in a single location or distributed amongst billions of participants. This is the case for example with federated learning (McMahan et al., 2017). Prior algorithms also assume that all parties are collectively training a single model with a fixed architecture. These requirements are often too restrictive in practice. For instance, a hospital may want to improve a medical diagnosis for a patient using data and models from other hospitals. In this case, the data is stored in multiple locations, and there are only a few parties collaborating. Further, each party may also want to train models with different architectures that best serve their own priorities.

We propose a new strategy that lets fewer heterogeneous parties learn from each other *collaboratively*, enabling each party to improve their own local models while protecting the confidentiality and privacy of their data. We call this *Confidential and Private Collaborative* (CaPC) learning.

Our strategy improves on confidential inference (Boemer, 2020) and PATE, the private aggregation of teacher ensembles (Papernot et al., 2017). Through structured applications of these two techniques, we design a strategy for inference that enables participants to operate an ensemble of heterogeneous models, i.e. the teachers, without having to explicitly join each party's data or teacher model at a single location. This also gives each party control at inference, because inference requires the agreement and participation of each party. In addition, our strategy provides measurable confidentiality and privacy guarantees, which we formally prove. We use the running example of a network of hospitals to illustrate our approach. The hospitals participating in CaPC protocol need guarantees on both *confidentiality* (i.e., data from a hospital can only be read by said hospital) and *privacy* (i.e., no hospital can infer private information about other hospitals' data by observing their predictions).

First, one hospital queries all the other parties over *homomorphic encryption* (HE), asking them to label an encrypted input using their own teacher models. This can prevent the other hospitals from reading the input (Boemer et al., 2019), an improvement over PATE, and allows the answering hospitals to provide a prediction to the querying hospital without sharing their teacher models.

The answering hospitals use *multi-party computation* (MPC) to compute an aggregated label, and add noise during the aggregation to obtain differential privacy guarantees (Dwork et al., 2014). This is achieved by a *privacy guardian* (PG), which then relays the aggregated label to the querying hospital. The PG only needs to be semi-trusted: we operate under the *honest-but-curious* assumption. The use of MPC ensures that the PG cannot decipher each teacher model's individual prediction, and the noise added via noisy argmax mechanism gives differential privacy even when there are *few participants*.

This is a significant advantage over prior decentralized approaches like federated learning, which require billions of participants to achieve differential privacy, because the sensitivity of the histogram used in our aggregation is lower than that of the gradients aggregated in federated learning. Unlike our approach, prior efforts involving few participants thus had to prioritize model utility over privacy and only guarantee confidentiality (Sheller et al., 2020).

Finally, the querying hospital can learn from this confidential and private label to improve their local model. Since the shared information is a label rather than a gradient, as used by federated learning, CaPC participants do not need to share a common model architecture; in fact, their *architectures can vary* throughout the participation in the protocol. This favors model development to a degree which is not possible in prior efforts such as federated learning.

We show how participants can instantiate various forms of *active and online learning* with the labels returned by our protocol: each party participating in the CaPC protocol may (a) identify deficiencies of its model throughout its deployment and (b) finetune the model with labels obtained by interacting with other parties. Intuitively, we achieve the analog of a doctor querying colleagues for a second opinion on a difficult diagnostic, without having to reveal the patient's medical condition. This protocol leads to improvements in both the accuracy and fairness (when there is a skew in the data distribution of each participating hospital) of model predictions for each of the CaPC participants.

To summarize, our contributions are the following:

- We introduce CaPC learning: a confidential and private collaborative learning platform that provides both confidentiality and privacy while remaining agnostic to ML techniques.
- Through a structured application of homomorphic encryption, secure MPC, and private aggregation, we design a protocol for CaPC. We use two-party deep learning inference and design an implementation of the noisy argmax mechanism with garbled circuits.
- Our experiments on SVHN and CIFAR10 demonstrate that CaPC enables participants to collaborate and improve the utility of their models, even in the heterogeneous setting where the architectures of their local models differ, and when there are only a few participants.
- Further, when the distribution of data drifts across participating parties, we show that CaPC significantly improves fairness metrics because querying parties benefit from knowledge learned by other parties on different data distributions, which is distilled in their predictions.
- We release the source code for reproducing all our experiments.

## 2 BACKGROUND

Before introducing CaPC, we first go over elements of cryptography and differential privacy that are required to understand it. Detailed treatment of these topics can be found in Appendices A and B.

### 2.1 CRYPTOGRAPHIC PRELIMINARIES FOR CONFIDENTIALITY

The main cryptographic tool used in CaPC is secure multi-party computation (MPC) (Yao, 1986). MPC allows a set of distrusting parties to jointly evaluate a function on their input without revealing anything beyond the output. In general, most practical MPC protocols can be classified into two categories: 1) generic MPC protocols that can compute any function with the above security goal (Malkhi et al., 2004); and 2) specialized MPC protocols that can be used to compute only selected functions (e.g., private set intersection (Pinkas et al., 2020), secure machine learning (Mohassel & Zhang, 2017)). Although specialized MPC protocols are less general, they are often more efficient in execution time. Protocols in both categories use similar cryptographic building blocks, including (fully) homomorphic encryption (Gentry, 2009), secret sharing (Shamir, 1979), oblivious transfer (Rabin, 2005), garbled circuits (Yao, 1986). To understand our protocol, it is not necessary to know all details about these cryptographic building blocks and thus we describe them in Appendix A.1. Our work uses these cryptographic preliminaries for secure computation at prediction time, unlike recent approaches, which explore new methods to achieving confidentiality at training time (Huang et al., 2020a;b).

The cryptographic protocol designed in this paper uses a specialized MPC protocol for securely evaluating a private ML model on private data, and a generic two-party computation protocol to compute an argmax in different forms. For the generic two-party computation, we use a classical Yao's

garbled-circuit protocol that can compute any function in Boolean circuit. For secure classification of neural networks, our protocol design is flexible to work with most existing protocols (Boemer et al., 2020; 2019; Gilad-Bachrach et al., 2016; Mishra et al., 2020). Most existing protocols are different in how they handle linear layers (e.g. convolution) and non-linear layers (e.g. ReLU). For instance, one can perform all computations using a fully homomorphic encryption scheme resulting in low communication but very high computation, or using classical MPC techniques with more communication but less computation. Other works (Juvekar et al., 2018) use a hybrid of both and thus enjoy further improvement in performance (Mishra et al., 2020). We discuss it in more details in Appendix A.2.

## 2.2 DIFFERENTIAL PRIVACY

Differential privacy is the established framework for measuring the privacy leakage of a randomized algorithm (Dwork et al., 2006). In the context of machine learning, it requires the training algorithm to produce statistically indistinguishable outputs on any pair of datasets that only differ by one data point. This implies that an adversary observing the outputs of the training algorithm (e.g., the model's parameters, or its predictions) can improve its guess at most by a bounded probability when inferring properties of the training data points. Formally, we have the following definition.

**Definition 1** (Differential Privacy). *A randomized mechanism $\mathcal{M}$ with domain $\mathcal{D}$ and range $\mathcal{R}$ satisfies $(\varepsilon, \delta)$-differential privacy if for any subset $\mathcal{S} \subseteq \mathcal{R}$ and any adjacent datasets $d, d' \in \mathcal{D}$, i.e. $\|d - d'\|_1 \leq 1$, the following inequality holds:*

$$\Pr\left[\mathcal{M}(d) \in \mathcal{S}\right] \leq e^{\varepsilon}\Pr\left[\mathcal{M}(d') \in \mathcal{S}\right] + \delta \tag{1}$$

In our work, we obtain differential privacy by post-processing the outputs of an ensemble of models with the noisy argmax mechanism of Dwork et al. (2014) (for more details on differential privacy, please refer to Appendix B), à la PATE (Papernot et al., 2017). We apply the improved analysis of PATE (Papernot et al., 2018) to compute the privacy guarantees obtained (i.e., a bound on $\varepsilon$). Our technique differs from PATE in that each of the teacher models is trained by different parties whereas PATE assumes a centralized learning setting where all of the training and inference is performed by a single party. Note that our technique is used at inference time, which differs from recent works in differential privacy that compare neuron pruning during training with mechanisms satisfying differential privacy (Huang et al., 2020c). We use cryptography to securely decentralize computations.

## 3 THE CAPC PROTOCOL

We now introduce our protocol for achieving both confidentiality and privacy in collaborative (CaPC) learning. To do so, we formalize and generalize our example of collaborating hospitals from Section 1.

### 3.1 PROBLEM DESCRIPTION

A small number of parties $\{\mathcal{P}_i\}_{i \in [1,K]}$, each holding a private dataset $\mathcal{D}_i = \{(x_j, y_j \text{ or } \varnothing)_{j \in [1,N_i]}\}$ and capable of fitting a predictive model $\mathcal{M}_i$ to it, wish to improve the utility of their individual models via collaboration. Due to the private nature of the datasets in question, they cannot directly share data or by-products of data (e.g., model weights) with each other. Instead, they will collaborate by querying each other for labels of the inputs about which they are uncertain. In the active learning paradigm, one party $\mathcal{P}_{i_*}$ poses queries in the form of data samples $x$ and all the other parties $\{\mathcal{P}_i\}_{i \neq i_*}$ together provide answers in the form of predicted labels $\hat{y}$. Each model $\{\mathcal{M}_i\}_{i \in [1,K]}$ can be exploited in both the querying phase and the answering phase, with the querying party alternating between different participants $\{\mathcal{P}_i\}_{i \in [1,K]}$ in the protocol.

**Threat Model.** To obtain the strong confidentiality and privacy guarantees that we described, we require a semi-trusted third party called the privacy guardian (PG). We assume that the PG does not collude with any party and that the adversary can corrupt any subset of $C$ parties $\{\mathcal{P}_i\}_{i \in [1,C]}$. When more than one party gets corrupted, this has no impact on the confidentiality guarantee, but the privacy budget obtained $\epsilon$ will degrade by a factor proportional to $C$ because the sensitivity of the aggregation mechanism increases (see Section 3.3). We work in the honest-but-curious setting, a commonly adopted assumption in cryptography which requires the adversary to follow the protocol description correctly but will try to infer information from the protocol transcript.

## 3.2 CaPC Protocol Description

Our protocol introduces a novel formulation of the private aggregation of teachers, which implements two-party confidential inference and secret sharing to improve upon the work of Papernot et al. (2017) and guarantee confidentiality. Recall that the querying party $P_{i_*}$ initiates the protocol by sending an encrypted input $x$ to all answering parties $\mathcal{P}_i, i \neq i_*$. We use $sk$ and $pk$ to denote the secret and public keys owned by party $\mathcal{P}_{i_*}$. The proposed protocol consists of the following steps:

1. For each $i \neq i_*$, $\mathcal{P}_i$ (with model parameters $\mathcal{M}_i$ as its input) and $\mathcal{P}_{i_*}$ (with $x, sk, pk$ as its input) run a secure two-party protocol. As the outcome, $\mathcal{P}_i$ obtains $\hat{s}_i$ and $\mathcal{P}_{i*}$ obtains $s_i$ such that $s_i + \hat{s}_i = \mathtt{OneHot}(\arg\max(r_i))$ where $r_i$ are the predicted logits.
   This step could be achieved by the following:
   a) $\mathcal{P}_{i_*}$ and $\mathcal{P}_i$ run a secure two-party ML classification protocol such that $\mathcal{P}_{i_*}$ learns nothing while $\mathcal{P}_i$ learns $\mathsf{Enc}_{pk}(r_i)$, where $r_i$ are the predicted logits.
   b) $\mathcal{P}_i$ generates a random vector $\hat{r}_i$, performs the following computation on the encrypted data $\mathsf{Enc}_{pk}(r_i) - \mathsf{Enc}_{pk}(\hat{r}_i) = \mathsf{Enc}_{pk}(r_i - \hat{r}_i)$, and sends the encrypted difference to $\mathcal{P}_{i_*}$, who decrypts and obtains $(r_i - \hat{r}_i)$.
   c) $\mathcal{P}_i$ (with $\hat{r}_i$ as input) and $\mathcal{P}_{i_*}$ (with $r_i - \hat{r}_i$ as input) engage in Yao's two-party garbled-circuit protocol to obtain vector $s_i$ for $\mathcal{P}_{i_*}$ and vector $\hat{s}_i$ for $\mathcal{P}_i$, such that $s_i + \hat{s}_i = \mathtt{OneHot}(\arg\max(r_i))$.
2. $\mathcal{P}_i$ sends $\hat{s}_i$ to the PG. The PG computes $\hat{s} = \sum_{i \neq i_*} \hat{s}_i + \mathrm{DPNoise}(\epsilon)$, where $\mathrm{DPNoise}()$ is element-wise Laplacian or Gaussian noise whose variance is calibrated to obtain a desired differential privacy guarantee $\varepsilon$; whereas $\mathcal{P}_{i_*}$ computes $s = \sum_{i \neq i_*} s_i$.
3. The PG and $P_{i_*}$ engage in Yao's two-party garbled-circuit protocol for computing the argmax: $\mathcal{P}_{i_*}$ gets $\arg\max(\hat{s} + s)$ and the PG gets nothing.

Next, we elaborate on the confidentiality and privacy guarantees achieved by CaPC.

## 3.3 Confidentiality and Differential Privacy Guarantees

**Confidentiality Analysis.** We prove in Appendix E that the above protocol reveals nothing to $\mathcal{P}_i$ or the PG and only reveals the final noisy results to $P_{i_*}$. The protocol is secure against a semi-honest adversary corrupting any subset of parties. Intuitively, the proof can be easily derived based on the security of the underlying components, including two-party classification protocol, secret sharing, and Yao's garbled circuit protocol. As discussed in Section 4.1 and Appendix A.1, for secret sharing of unbounded integers, we need to make sure the random padding is picked from a domain much larger than the maximum possible value being shared. Given the above, a corrupted $\mathcal{P}_{i_*}$ cannot learn anything about $\mathcal{M}_i$ of the honest party due to the confidentiality guarantee of the secure classification protocol; similarly, the confidentiality of $x$ against corrupted $\mathcal{P}_i$ is also protected. Intermediate values are all secretly shared (and only recovered within garbled circuits) so they are not visible to any party.

**Differential Privacy Analysis.** Here, any potential privacy leakage in terms of differential privacy is incurred by the answering parties $\{\mathcal{P}_i\}_{i \neq i_*}$ for their datasets $\{\mathcal{D}_i\}_{i \neq i_*}$, because these parties share the predictions of their models. Before sharing these predictions to $\mathcal{P}_{i_*}$, we follow the PATE protocol: we compute the histogram of label counts $\hat{y}$, then add Laplacian or Gaussian noise using a sensitivity of 1, and finally return the argmax of $\hat{y}_\sigma$ to $\mathcal{P}_{i_*}$. Since $\mathcal{P}_{i_*}$ only sees this noisily aggregated label, both the data-dependent and data-independent differential privacy analysis of PATE apply to $\mathcal{P}_{i_*}$ (Papernot et al., 2017; 2018). Thus, when there are enough parties with high consensus, we can obtain a tighter bound on the privacy budget $\epsilon$ as the true plurality will more likely be returned (refer to Appendix B for more details on how this is achieved in PATE). This setup assumes that only one answering party can be corrupted. If instead $C$ parties are corrupted, the sensitivity of the noisy aggregation mechanism will be scaled by $C$ and the privacy guarantee will deteriorate. There is no privacy leakage to the PG; it does not receive any part of the predictions from $\{\mathcal{P}_i\}_{i \neq i_*}$.

## 4 Experiments

CaPC aims to improve the model utility of collaborating parties by providing them with new labelled data for training their respective local models. Since we designed the CaPC protocol with techniques

for confidentiality (i.e., confidential inference and secret sharing) and differential privacy (i.e., private aggregation), our experiments consider the following three major dimensions:

1. How well does collaboration improve the model utility of all participating parties?

2. What requirements are there to achieve privacy and how can these be relaxed under different circumstances? What is the trade-off between the privacy and utility provided by CaPC?

3. What is the resulting computational cost for ensuring confidentiality?

## 4.1 IMPLEMENTATION

We use the HE-transformer library with MPC (MP2ML) by Boemer (2020) in step **1a** of our protocol for confidential two-party deep learning inference. To make our protocol flexible to any private inference library, not just those that return the label predicted by the model (HE-transformer only returns logits), we incorporate steps **1b** and **1c** of the protocol outside of the private inference library. The EMP toolkit (Wang et al., 2016) for generic two-party computation is used to compute the operations including $argmax$ and $sum$ via the garbled circuits. To secret share the encrypted values, we first convert them into integers over a prime field according to the CKKS parameters, and then perform secret sharing on that domain to obtain perfect secret sharing. We use the single largest logit value for each $\mathcal{M}_i$ obtained on its training set $\mathcal{D}_i$ in plain text to calculate the necessary noise.

## 4.2 EVALUATION SETUP

**Collaboration.** We use the following for experiments unless otherwise noted. We uniformly sample from the training set in use[2], without replacement, to create disjoint partitions, $\mathcal{D}_i$, of equal size and identical data distribution for each party. We select $K = 50$ and $K = 250$ as the number of parties for CIFAR10 and SVHN, respectively (the number is larger for SVHN because we have more data). We select $Q = 3$ querying parties, $\mathcal{P}_{i_*}$, and similarly divide part of the test set into $Q$ separate private pools for each $\mathcal{P}_{i_*}$ to select queries, until their privacy budget of $\epsilon$ is reached (using Gaussian noise with $\sigma = 40$ on SVHN and 7 on CIFAR10). We are left with $1,000$ and $16,032$ evaluation data points from the test set of CIFAR10 and SVHN, respectively. We fix $\epsilon = 2$ and 20 for SVHN and CIFAR10, respectively (which leads to $\approx 550$ queries per party), and report accuracy on the evaluation set. Querying models are retrained on their $\mathcal{D}_i$ plus the newly labelled data; the difference in accuracies is their accuracy improvement.

We use shallower variants of VGG, namely VGG-5 and VGG-7 for CIFAR10 and SVHN, respectively, to accommodate the small size of each party's private dataset. We instantiate VGG-7 with 6 convolutional layers and one final fully-connected layer, thus there are 7 functional layers overall. Similarly, VGG-5 has 4 convolutional layers followed by a fully connected layer. The ResNet-10 architecture starts with a single convolutional layer, followed by 4 basic blocks with 2 convolutional layers in each block, and ends with a fully-connected layer, giving 10 functional layers in total. The ResNet-8 architecture that we use excludes the last basic block and increases the number of neurons in the last (fully-connected) layer. We present more details on architectures in Appendix F.2.

We first train local models for all parties using their non-overlapping private datasets. Next, we run the CaPC protocol to generate query-answer pairs for each querying party. Finally, we retrain the local model of each querying party using the combination of their original private dataset and the newly obtained query-answer pairs. We report the mean accuracy and class-specific accuracy averaged over 5 runs for all retrained models, where each uses a different random seed.

**Heterogeneity and Data Skew.** Where noted, our heterogeneous experiments (recall that this is a newly applicable setting that CaPC enables) use VGG-7, ResNet-8 and ResNet-10 architectures for $\frac{K}{3}$ parties, each. One model of each architecture is used for each of $Q = 3$ querying parties. Our data skew experiments use $80\%$ less data samples for the classes 'horse', 'ship', and 'truck' on CIFAR10 and $90\%$ less data for the classes 1 and 2 on SVHN. In turn, unfair ML algorithms perform worse on these specific classes, leading to worse *balanced accuracy* (see Appendix D). We adopt balanced accuracy instead of other fairness metrics because the datasets we use have no sensitive attributes, making them inapplicable. We employ margin, entropy, and greedy k-center active learning strategies

---

[2]For the SVHN dataset, we combine its original training set and extra set to get a larger training set.

(described in Appendix C) to encourage ML algorithms to sample more queries from regimes that have been underrepresented and to improve their fairness performance.

### 4.3 COLLABORATION ANALYSIS

We first investigate the benefits of collaboration for improving each party's model performance in several different settings, namely: homogeneous and heterogeneous model architectures across querying and answering parties, and uniform and non-uniform data sampling for training data. From these experiments, we observe: increased accuracy in both homogeneous settings and heterogeneous settings to all model architectures (Section 4.3.1) and improved balanced accuracy when there is data skew between parties, i.e., non-uniform private data (Section 4.3.2).

#### 4.3.1 UNIFORMLY SAMPLED PRIVATE DATA

The first setting we consider is a uniform distribution of data amongst the parties—there is no data drift among parties. Our set up for the uniform data distribution experiments is detailed in Section 4.2. We evaluate the per-class and overall accuracy before and after CaPC in both homogeneous and heterogeneous settings on the CIFAR10 and SVHN datasets.

In Figure 2, we see there is a consistent increase in accuracy for each class and overall in terms of mean accuracy across all parties on the test sets. We observe these improvements in both the homogeneous and heterogeneous settings for both datasets tested. As demonstrated in Figure 2, there is a greater climb in mean accuracy for the heterogeneous setting than the homogeneous setting on SVHN. Figures 5, 6, and 7 provide a breakdown of the benefits obtained by each querying party. We can see from these figures that all querying parties observe an increase in overall accuracy in heterogeneous and homogeneous settings with both datasets; additionally, the jump in accuracy is largely constant between different model architectures. In only $6.67\%$ of all cases were any *class-specific* accuracies degraded, but they still showed a net increase in overall model accuracy.

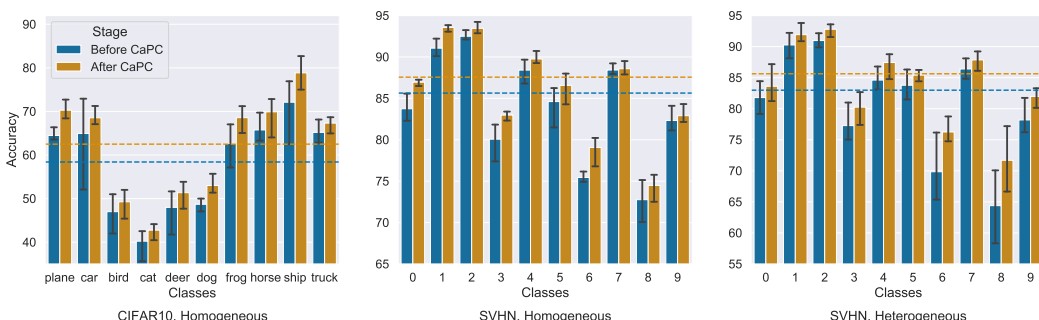

Figure 2: **Using CaPC to improve model performance.** *Dashed lines represent mean accuracy.* With homogeneous models, we observe a mean increase of $4.09$ and of $1.92$ percentage points on CIFAR10 and SVHN, respectively, and an increase of $2.64$ with heterogeneous models; each party still sees improvements despite differing model architectures (see Figure 7 in Appendix F).

#### 4.3.2 NON-UNIFORMLY SAMPLED PRIVATE DATA

In this section, we focus our analysis on two types of data skew between parties: varying size of data per class and total size of data provided; the setup is described in Section 4.2. To analyze data skew, we explore the balanced accuracy (which measures mean recall on a per-class basis, see Appendix D). We use balanced accuracy in order to investigate aggregate fairness gains offered by CaPC. Random sampling from non-uniform distributions leads to certain pitfalls: e.g., underrepresented classes are not specifically targeted in sampling. Thus, we additionally utilize active learning techniques, namely entropy, margin, and greedy-k-center (see Definitions 6-8 in Appendix C), and analyze balanced accuracy with each strategy.

In Figure 3, we see that CaPC has a significant impact on the balanced accuracy when there is data skew between the private data of participating parties. Even random sampling can drastically improve balanced accuracy. Leveraging active learning techniques, we can achieve additional benefits in

balanced accuracy. In particular, we observe that entropy and margin sampling achieves the greatest improvement over random sampling in per-class accuracy for the less represented classes 'horse', 'ship', and 'truck' on CIFAR10 and classes 1 and 2 on SVHN. These enhancements can be explained by the underlying mechanisms of margin and entropy sampling because the less-represented classes have a higher margin/entropy; the queries per class for each method are shown in Figure 9. Through these experiments, we show that in data skew settings, the CaPC protocol can significantly improve the fair performance of models (as measured by balanced accuracy), especially when combined with active learning techniques. Note that we see similar trends with (normal) accuracy as well.

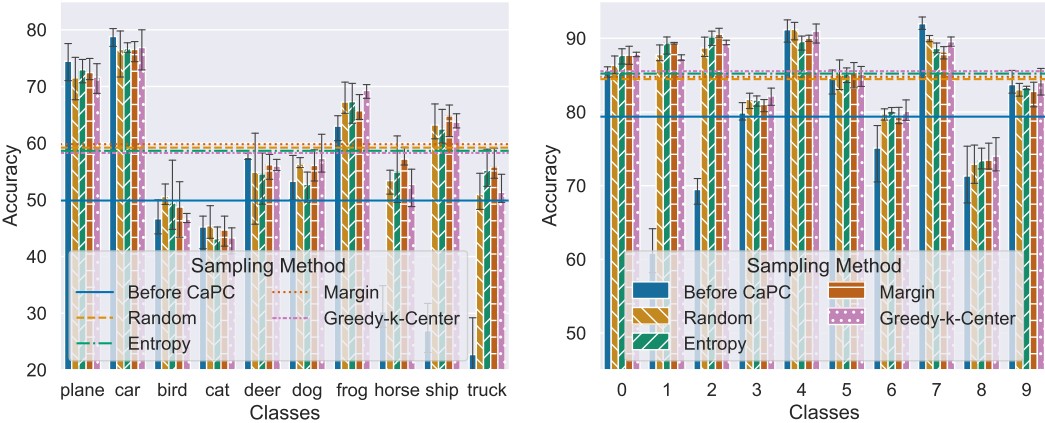

Figure 3: **Using CaPC with active learning to improve balanced accuracy under non-uniform data distribution.** *Dashed lines are balanced accuracy (BA).* We observe that all sampling strategies significantly improve BA and the best active learning scheme can improve BA by a total of $9.94$ percentage-points (an additional $0.8$ percentage points over Random sampling) on CIFAR10 (left) and a total of $5.67$ percentage-points (an additional $0.38$) on SVHN (right).

## 4.4 PRIVACY VERSUS UTILITY

We now study the trade-off between privacy and utility of our obtained models. Recall that we add Gaussian (or Laplacian) noise to the aggregate of predicted labels of all parties. Under the uniform setting, we choose the standard deviation $\sigma$ by performing a (random) grid search and choosing the highest noise before a significant loss in accuracy is observed. In doing so, each query uses minimal $\varepsilon$ while maximizing utility. Figure 11 in Appendix F shows a sample plot for $K = 250$ models. For more details on how $\varepsilon$ is calculated, please refer to Appendix B.

As we increase the number of parties, we can issue more queries for a given privacy budget ($\varepsilon$) which leads to a higher accuracy gain. In Figure 4, we report the accuracy gain achieved using CaPC with various numbers of parties, $K$. With a fixed total dataset size, increasing the number of parties decreases their training data size, leading to worse performing models. These models see the largest benefit from CaPC but, importantly, we always see a net improvement across all values of $K$.

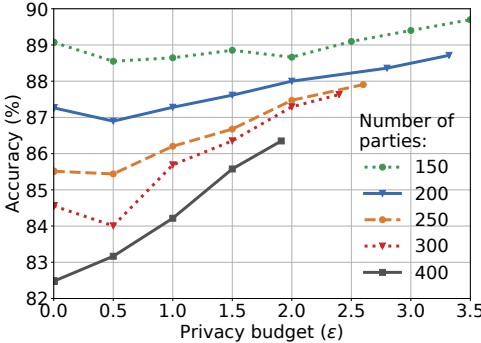

|  | **Number of parties** | | | | |
| --- | --- | --- | --- | --- | --- |
|  | 150 | 200 | 250 | 300 | 400 |
| **Accuracy gain (%)** | 0.62 | 1.45 | 2.39 | 3.07 | 3.87 |
| **Best $\varepsilon$** | 3.50 | 3.32 | 2.60 | 2.40 | 1.91 |

Figure 4: **Accuracy gain for balanced SVHN using CaPC versus number of parties and privacy budget, $\varepsilon$.** With more parties, we can achieve a higher accuracy gain at a smaller bound on $\varepsilon$.

## 4.5 COMPUTATIONAL COSTS OF CONFIDENTIALITY

The incorporation of confidentiality in CaPC increases computational costs. We segment the analysis of computational overhead of CaPC into three parts corresponding to sequential steps in the protocol: (1) inference, (2) secret sharing between each querying and answering party, and (3) secret sharing between the querying party and the PG. Each of these steps is analyzed in terms of the wall-clock time (in seconds). We use the default encryption setting in HE-transformer and vary the modulus range, $N$, which denotes the max value of a given plain text number to increase the maximum security level possible. HE-transformer only supports inference on CPUs and is used in step (1).

Step (1) with neural network inference using MPC incurs the highest CPU and network costs (see Table 1 and Figure 13 in Appendix F). Even the base level of security increases computational cost by 100X, and high security levels see increases up to 1000X, in comparison to the non-encrypted inference on CPU. Compared to step (1), the rest of the CaPC protocol incurs a negligible overhead to perform secret sharing. Overall, CaPC incurs only a low additional cost over the underlying MP2ML framework, as shown in Figure 13, which enables applicability and scalability as these tools progress.

## 5 DISCUSSION AND CONCLUSIONS

CaPC is a secure and private protocol that protects both the confidentiality of test data and the privacy of training data, which are desired in applications like healthcare and finance. Our framework facilitates collaborative learning using heterogeneous model architectures and separate private datasets, even if the number of parties involved is small. It offers notable advantages over recent methods for learning with multiple participants, such as federated learning, which assumes training of a single fixed model architecture. CaPC does not assume a homogeneous model architecture and allows parties to separately and collaboratively train different models optimized for their own purposes. Federated learning also requires a large number of parties while CaPC provides gains in accuracy with significantly fewer participants, even in contexts where each party already has a model with high accuracy. Notably, CaPC incurs low overhead on top of underlying tools used for secure neural network inference.

Through our experiments, we also demonstrate that CaPC facilitates collaborative learning even when there exists non i.i.d (highly skewed) private data among parties. Our experiments show that CaPC improves on the fair performance of participating querying models as indicated by improvements in the balanced accuracy, a common fairness metric. Further, we observe a significant increase in per-class accuracy on less-represented classes on all datasets tested. Notably, CaPC is easily configured to leverage active learning techniques to achieve additional fairness improvement gains or to learn from other heterogeneous models trained with fairness techniques, e.g., with synthetic minority oversampling (Chawla et al., 2002). In future work, we look to analyzing the fairness implications of CaPC in contexts where there is discrimination over a private dataset's sensitive attributes, not just class labels. In these cases, other fairness metrics like equalized odds and equal opportunity (see Appendix D) can be explored.

We note some limitations of the proposed protocol. HE-transformer does not prevent leaking certain aspects of the model architecture, such as the type of non-linear activation functions and presence of MaxPooling layers. CaPC improves upon existing methods in terms of the necessary number of parties; however, it would be favorable to see this number decreased under 50 for better flexibility and applicability in practice.

In the face of this last limitation, when there are few physical parties, we can generate a larger number of virtual parties for CaPC, where each physical party subdivides their private dataset into disjoint partitions and trains multiple local models. This would allow CaPC to tolerate more noise injected during aggregation and provide better privacy guarantees. Note that each physical party could select queries using a dedicated strong model instead of the weak models used for answering queries in CaPC. This setting is desirable in cases where separate models are required within a single physical party, for example, in a multi-national organization with per-country models.

ACKNOWLEDGMENTS

We would like to acknowledge our sponsors, who support our research with financial and in-kind contributions: Microsoft, Intel, CIFAR through the Canada CIFAR AI Chair and AI catalyst programs, NFRF through an Exploration grant, and NSERC COHESA Strategic Alliance. Resources used in preparing this research were provided, in part, by the Province of Ontario, the Government of Canada through CIFAR, and companies sponsoring the Vector Institute www.vectorinstitute.ai/partners. Finally, we would like to thank members of CleverHans Lab for their feedback, especially: Tejumade Afonja, Varun Chandrasekaran, Stephan Rabanser, and Jonas Guan.

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

## A    MORE BACKGROUND ON CRYPTOGRAPHY

### A.1    CRYPTOGRAPHIC BUILDING BLOCKS

**Homomorphic encryption.** Homomorphic encryption defines an encryption scheme such that the encryption and decryption functions are homomorphic between plaintext and ciphertext spaces. Although it is known that fully homomorphic encryption can be constructed based on lattice-based assumptions, most applications only require a weaker version with bounded number of multiplications on each ciphertext. Schemes with this constraint are much more practical, including for example, BGV (Brakerski et al., 2014), CKKS (Cheon et al., 2017), etc.

**Secret sharing.** Secret sharing denotes a scheme in which a datum, the secret, is shared amongst a group of parties by dividing the secret into parts such that each party only has one part, or 'share' of the secret. The secret can only be recovered if a certain number of parties conspire to combine their shares. It is easy to construct secret sharing modulo a positive integer. If the application does not allow modular operation, one can still achieve statistically secure secret sharing by using random shares that are much larger than the secret being shared (Evans et al., 2011).

**Oblivious transfer.** Oblivious transfer involves two parties: the sending party and the receiving party. The sending party has two pieces of information, $s_0$ and $s_1$, and the receiver wants to receive $s_b$, where $b \in \{0, 1\}$, such that the sending party cannot learn $b$ and the receiving party cannot learn $s_{\neg b}$. In general, oblivious transfer requires public-key operations, however, it is possible to execute a large number of oblivious transfers with only a very small number of public-key operations based on oblivious transfer extension (Ishai et al., 2003).

**Garbled circuits.**   In Yao's garbled circuit protocol for two-party computation, each of the two parties assumes a role, that of garbler or that of evaluator. The function f on which to compute each of the two parties' inputs is described as a Boolean circuit. The garbler randomly generates aliases (termed labels) representing 0 and 1 in the Boolean circuit describing f and replaces the binary values with the generated labels for each wire in the circuit. At each gate in the circuit, which can be viewed as a truth table, the garbler uses the labels of each possible combination of inputs to encrypt the corresponding outputs, and permutes the rows of the truth table. The garbler then uses the generated labels for 0 and 1 to encode their own input data and sends these labels and the garbled Boolean circuit to the evaluator. The evaluator now converts their binary input data to the corresponding labels through a 1-2 oblivious transfer protocol with the garbler. After receiving the labels for their input, the evaluator evaluates the garbled circuit by trying to decrypt each row in the permutable truth tables at each gate using the input labels; only one row will be decryptable at each gate, which is the output label for the outgoing wire from the gate. The evaluator eventually finishes evaluating the garbled circuit and obtains the label for the output of the function f computed on the garbler's and the evaluator's input. The garbler then must provide the true value for the output label so that both parties can get the output.

## A.2   PROTECTING CONFIDENTIALITY USING MPC

Neural networks present a challenge to secure multi-party computation protocols due to their unique structure and exploitative combination of linear computations and non-linear activation functions. Cryptographic inference with neural networks can be considered in two party computation case in which one party has confidential input for which they wish to obtain output from a model and the other party stores the model; in many cases the party storing the model also wishes that the model remains secure.

Confidential learning and inference with neural networks typically uses homomorphic encryption (HE) or secure multi-party computation (MPC) methods. Many libraries support pure HE or MPC protocols for secure inference of neural networks; a comprehensive list can be viewed in (Boemer et al., 2020). Notably, libraries such as nGraph-HE (Boemer et al., 2019) and CryptoNets (Gilad-Bachrach et al., 2016) provide pure homomorphic encryption solutions to secure neural network inference. nGraph-HE, an extension of graph compiler nGraph, allows secure inference of DNNs through linear computations at each layer using CKKS homomorphic encryption scheme (Cheon et al., 2017; Boemer et al., 2019). CryptoNets similarly permit confidential neural network inference using another leveled homomorphic encryption scheme, YASHE' (Gilad-Bachrach et al., 2016). On the other hand, several libraries employing primarily MPC methods in secure NN inference frameworks rely on ABY, a tool providing support for common non-polynomial activation functions in NNs through use of both Yao's GC and GMW.

In DL contexts, while pure homomorphic encryption methods maintain model security, their failure to support common non-polynomial activation functions leads to leaking of pre-activation values (feature maps at hidden layers). Tools that use solely MPC protocols avoid leaking pre-activation values as they can guarantee data confidentiality on non-polynomial activation functions but may compromise the security of the model architecture by leaking activation functions or model structure.

Recent works on secure NN inference propose hybrid protocols that combine homomorphic encryption schemes, and MPC methods to build frameworks that try to reduce leakages common in pure HE and MPC protocols. Among recent works that use hybrid protocols and do not rely on trusted third parties are Gazelle (Juvekar et al., 2018), Delphi (Mishra et al., 2020), and MP2ML (Boemer et al., 2020).

Gazelle, Delphi and MP2ML largely support non-polynomial activation functions encountered in convolutional neural networks, such as maximum pooling and rectified linear unit (ReLU) operations. Gazelle introduced several improvements over previous methods for secure NN inference primarily relating to latency and confidentiality. In particular, Gazelle framework provides homomorphic encryption libraries with low latency implementations of algorithms for single instruction multiple data (SIMD) operations, ciphertext permutation, and homomorphic matrix and convolutional operations, pertinent to convolutional neural networks. Gazelle utilizes kernel methods to evaluate homomorphic operations for linear components of networks, garbled circuits to compute non-linear activation functions confidentially and additive secret sharing to quickly switch between these cryptographic protocols. Delphi builds on Gazelle, optimizing computation of both linear and non-linear com-

putations in CNNs by secret sharing model weights in the pre-processing stage to speed up linear computations later, and approximating certain activation functions such as ReLU with polynomials. MP2ML employs nGraph-HE for homomorphic encryption and ABY framework for evaluation of non-linear functions using garbled circuits.

# B    MORE BACKGROUND ON DIFFERENTIAL PRIVACY

One of the compelling properties of differential privacy is that it permits the analysis and control of cumulative privacy cost over multiple consecutive computations. For instance, strong composition theorem (Dwork et al., 2010) gives a tight estimate of the privacy cost associated with a sequence of adaptive mechanisms $\{\mathcal{M}_i\}_{i \in I}$.

**Theorem 1** (Strong Composition). *For $\varepsilon, \delta, \delta' \geq 0$, the class of $(\varepsilon, \delta)$-differentially private mechanisms satisfies $(\varepsilon', k\delta + \delta')$-differential privacy under $k$-fold adaptive composition for:*

$$\varepsilon' = \varepsilon\sqrt{2k \log(1/\delta')} + k\varepsilon(e^\varepsilon - 1) \tag{2}$$

To facilitate the evaluation of privacy leakage resulted by a randomized mechanism $\mathcal{M}$, it is helpful to explicitly define its corresponding privacy loss $c_\mathcal{M}$ and privacy loss random variable $C_\mathcal{M}$. Particularly, the fact that $\mathcal{M}$ is $(\varepsilon, \delta)$-differentially private is equivalent to a certain tail bound on $C_\mathcal{M}$.

**Definition 2** (Privacy Loss). *Given a pair of adjacent datasets $d, d' \in \mathcal{D}$ and an auxiliary input $aux$, the privacy loss $c_\mathcal{M}$ of a randomized mechanism $\mathcal{M}$ evaluated at an outcome $o \in \mathcal{R}$ is defined as:*

$$c_\mathcal{M}(o \mid aux, d, d') \triangleq \log \frac{\Pr[\mathcal{M}(aux, d) = o]}{\Pr[\mathcal{M}(aux, d') = o]} \tag{3}$$

*For an outcome $o \in \mathcal{R}$ sampled from $\mathcal{M}(d)$, $C_\mathcal{M}(aux, d, d')$ takes the value $c_\mathcal{M}(o \mid aux, d, d')$.*

Based on the definition of privacy loss, Abadi et al. (Abadi et al., 2016) introduced the moments accountant to track higher-order moments of privacy loss random variable and achieved even tighter privacy bounds for $k$-fold adaptive mechanisms.

**Definition 3** (Moments Accountant). *Given any adjacent datasets $d, d' \in \mathcal{D}$ and any auxiliary input $aux$, the moments accountant of a randomized mechanism $\mathcal{M}$ is defined as:*

$$\alpha_\mathcal{M}(\lambda) \triangleq \max_{aux, d, d'} \alpha_\mathcal{M}(\lambda \mid aux, d, d') \tag{4}$$

*where $\alpha_\mathcal{M}(\lambda \mid aux, d, d') \triangleq \log \mathbb{E}[\exp(\lambda C_\mathcal{M}(aux, d, d'))]$ is obtained by taking the logarithm of the privacy loss random variable.*

As a natural relaxation to the conventional $(\varepsilon, \delta)$-differential privacy, Rényi differential privacy (RDP) (Mironov, 2017) provides a more convenient and accurate approach to estimating privacy loss under heterogeneous composition.

**Definition 4** (Rényi Divergence). *For two probability distributions $P$ and $Q$ defined over $\mathcal{R}$, the Rényi divergence of order $\lambda > 1$ between them is defined as:*

$$D_\lambda(P \mid\mid Q) \triangleq \frac{1}{\lambda - 1} \log \mathbb{E}_{x \sim Q}\left[(P(x)/Q(x))^\lambda\right] = \frac{1}{\lambda - 1} \log \mathbb{E}_{x \sim P}\left[(P(x)/Q(x))^{\lambda - 1}\right] \tag{5}$$

**Definition 5** (Rényi Differential Privacy). *A randomized mechanism $\mathcal{M}$ is said to satisfy $\varepsilon$-Rényi differential privacy of order $\lambda$, or $(\lambda, \varepsilon)$-RDP for short, if for any adjacent datasets $d, d' \in \mathcal{D}$:*

$$D_\lambda(\mathcal{M}(d) \mid\mid \mathcal{M}(d')) = \frac{1}{\lambda - 1} \log \mathbb{E}_{x \sim \mathcal{M}(d)}\left[\left(\frac{\Pr[\mathcal{M}(d) = x]}{\Pr[\mathcal{M}(d') = x]}\right)^{\lambda - 1}\right] \leq \varepsilon \tag{6}$$

**Theorem 2** (From RDP to DP). *If a randomized mechanism $\mathcal{M}$ guarantees $(\lambda, \varepsilon)$-RDP, then it also satisfies $(\varepsilon + \frac{\log(1/\delta)}{\lambda - 1}, \delta)$-differential privacy for any $\delta \in (0, 1)$.*

Building upon the moments accountant and RDP techniques, Private Aggregation of Teacher Ensembles (PATE) (Papernot et al., 2017) provides a flexible approach to training machine learning models with strong privacy guarantees. Precisely, rather than directly learning from labeled private

data, the model that gets released instead learns from unlabeled public data by querying a teacher ensemble for predicted labels. Models in the ensemble are themselves trained on disjoint partitions of the private dataset, while privacy guarantees are enabled by applying the Laplace mechanism to the ensemble's aggregated label counts. Coupled with data-dependent privacy analysis, PATE achieves a tighter estimate of the privacy loss associated with label queries, especially when the consensus among teacher models is strong. Given this motivation, the follow-up work of PATE (Papernot et al., 2018) further improves the privacy bound both by leveraging a more concentrated noise distribution to strengthen consensus and by rejecting queries that lack consensus.

## C    MORE BACKGROUND ON ACTIVE LEARNING

Active learning, sometimes referred to as query learning, exploits the intuition that machine learning algorithms will be able to learn more efficiently if they can actively select the data from which they learn. For certain supervised learning tasks, this insight is of particularly important implications, as labeled data rarely exists in abundance and data labeling can be very demanding (Settles, 2009).

In order to pick queries that will most likely contribute to model learning, various pool sampling methods have been proposed to estimate the informativeness of unlabeled samples. Uncertainty-based approaches (Lewis & Gale, 1994), such as margin sampling and entropy sampling, typically achieve a satisfactory trade-off between sample utility and computational efficiency. We also explore a core-set approach to active learning using greedy-k-center sampling (Sener & Savarese, 2017).

**Definition 6** (Margin Sampling (Scheffer et al., 2001)). *Given an unlabeled dataset $d$ and a classification model with conditional label distribution $P_\theta(y \mid x)$, margin sampling outputs the most informative sample:*

$$x^* = \arg\min_{x \in d} P_\theta(\hat{y}_1 \mid x) - P_\theta(\hat{y}_2 \mid x) \tag{7}$$

*where $\hat{y}_1$ and $\hat{y}_2$ stand for the most and second most probable labels for $x$, according to the model.*

**Definition 7** (Entropy Sampling). *Using the setting and notations in Definition 6, margin sampling can be generalized by using entropy (Shannon, 1948) as an uncertainty measure as follows:*

$$x^* = \arg\max_{x \in d} - \sum_i P_\theta(y_i \mid x) \log P_\theta(y_i \mid x) \tag{8}$$

*where $y_i$ ranges over all possible labels.*

**Definition 8** (Greedy-K-center Sampling). *We aim to solve the k-center problem defined by Farahani & Hekmatfar (2009), which is, intuitively, the problem of picking $k$ center points that minimize the largest distance between a data point and its nearest center. Formally, this goal is defined as*

$$\min_{\mathcal{S}:|\mathcal{S} \cup \mathcal{D}| \le k} \max_i \min_{j \in \mathcal{S} \cup \mathcal{D}} \Delta(\mathbf{x}_i, \mathbf{x}_j) \tag{9}$$

*where $\mathcal{D}$ is the current training set and $\mathcal{S}$ is our new chosen center points. This definition can can be solved greedily as shown in  (Sener & Savarese, 2017).*

## D    MORE BACKGROUND ON FAIRNESS

Due to the imbalance in sample quantity and learning complexity, machine learning models may have disparate predictive performance over different classes or demographic groups, resulting in unfair treatment of certain population. To better capture this phenomenon and introduce tractable countermeasures, various fairness-related criteria have been proposed, including balanced accuracy, demographic parity, equalized odds (Hardt et al., 2016), etc.

**Definition 9** (Balanced Accuracy). *Balanced accuracy captures model utility in terms of both accuracy and fairness. It is defined as the average of recall scores obtained on all classes.*

Among the criteria that aim to alleviate discrimination against certain protected attributes, equalized odds and equal opportunity Hardt et al. (2016) are of particular research interests.

**Definition 10** (Equalized Odds). *A machine learning model is said to guarantee equalized odds with respect to protected attribute $A$ and ground truth label $Y$ if its prediction $\hat{Y}$ and $A$ are conditionally independent given $Y$. In the case of binary random variables $A, Y, \hat{Y}$, this is equivalent to:*

$$\Pr\left[\hat{Y} = 1 \mid A = 0, Y = y\right] = \Pr\left[\hat{Y} = 1 \mid A = 1, Y = y\right], \quad y \in \{0, 1\} \tag{10}$$

*To put it another way, equalized odds requires the model to have equal true positive rates and equal false positive rates across the two demographic groups A = 0 and A = 1.*

**Definition 11** (Equal Opportunity). *Equal opportunity is a relaxation of equalized odds that requires non-discrimination only within a specific outcome group, often referred to as the advantaged group. Using previous notations, the binary case with advantaged group $Y = 1$ is equivalent to:*

$$\Pr\left[\hat{Y} = 1 \,|\, A = 0, Y = 1\right] = \Pr\left[\hat{Y} = 1 \,|\, A = 1, Y = 1\right] \tag{11}$$

## E    PROOF OF CONFIDENTIALITY

Here we prove that our protocol described in the main body does not reveal anything except the final noised result to $P_{i_*}$. In can be proven in the standard real-world ideal-world paradigm, where the ideal functionality takes inputs from all parties and sends the final results to $P_{i_*}$. We use $\mathcal{A}$ to denote the set of corrupted parties. Below, we describe the simulator (namely $\mathcal{S}$). The simulator strategy depends on if $i_*$ is corrupted.

If $i_* \in \mathcal{A}$, our simulator works as below:

   1.a) The simulator simulates what honest parties would do.

   1.b) For each $i \notin \mathcal{A}$, $\mathcal{S}$ sends fresh encryption of a random $r_i$ to $P_{i_*}$.

   1.c) For each $i \notin \mathcal{A}$, $\mathcal{S}$ sends random $s_i$ to $P_{i_*}$ on be half of the 2PC functionality between $P_i$ and $P_{i_*}$.

   2-3 $\mathcal{S}$ sends the output of the whole computation to $P_{i_*}$ on behalf of the 2PC functionality between PG and $P_{i_*}$

If $i_* \notin \mathcal{A}$, our simulator works as below:

   1.a) If $i_* \notin \mathcal{A}$, for each $i \in \mathcal{A}$, $\mathcal{S}$ computes a fresh encryption of zero and sends it to $P_i$ on behalf of $P_{i_*}$.

   1.b) The simulator simulates what honest parties would do.

   1.c) For each $i \in \mathcal{A}$, $\mathcal{S}$ sends random $\hat{s}_i$ to $P_i$ on behalf of the 2PC functionality between $P_i$ and $P_{i_*}$.

   2-3 The simulator simulates what honest parties would do.

Assuming that the underlying encryption scheme is CPA secure and that 2PC protocols used in step 1, 2 and 3 are secure with respect to standard definitions (i.e., reveals nothing beyond the outputs), our simulation itself is perfect.

## F    DETAILS ON EXPERIMENTAL SETUP

### F.1    MNIST AND FASHION-MNIST

We use the same setup as for CIFAR10 and SVHN datasets with the following adjustments. We select $K = 250$ as the default number of parties. For the imbalanced classes we select classes 1 and 2 for MNIST as well as Trouser and Pullover for Fashion-MNIST. We use the Gaussian noise with $\sigma = 40$ (similarly to SVHN). We are left with $1,000$ evaluation data points from the test set (similarly to CIFAR10). We fix the default value of $\epsilon = 2.35$ for MNIST and $\epsilon = 3.89$ for Fashion-MNIST. We use a variant of the LeNet architecture.

### F.2    DETAILS ON ARCHITECTURES

To train the private models on subsets of datasets, we downsize the standard architectures, such as VGG-16 or ResNet-18. Below is the detailed list of layers in each of the architectures used (generated using *torchsummary*). The diagram for ResNet-10 also includes skip connections and convolutional layers for adjusting the sizes of feature maps.

```
VGG-7 for SVHN:
----------------------------------------------------------------
       Layer type            Output Shape            Param #
================================================================
         Conv2d-1         [-1, 64, 32, 32]             1,728
    BatchNorm2d-2         [-1, 64, 32, 32]               128
           ReLU-3         [-1, 64, 32, 32]                 0
      MaxPool2d-4         [-1, 64, 16, 16]                 0
         Conv2d-5        [-1, 128, 16, 16]            73,728
    BatchNorm2d-6        [-1, 128, 16, 16]               256
           ReLU-7        [-1, 128, 16, 16]                 0
      MaxPool2d-8          [-1, 128, 8, 8]                 0
         Conv2d-9          [-1, 256, 8, 8]           294,912
   BatchNorm2d-10          [-1, 256, 8, 8]               512
          ReLU-11          [-1, 256, 8, 8]                 0
        Conv2d-12          [-1, 256, 8, 8]           589,824
   BatchNorm2d-13          [-1, 256, 8, 8]               512
          ReLU-14          [-1, 256, 8, 8]                 0
     MaxPool2d-15          [-1, 256, 4, 4]                 0
        Conv2d-16          [-1, 512, 4, 4]         1,179,648
   BatchNorm2d-17          [-1, 512, 4, 4]             1,024
          ReLU-18          [-1, 512, 4, 4]                 0
        Conv2d-19          [-1, 512, 4, 4]         2,359,296
   BatchNorm2d-20          [-1, 512, 4, 4]             1,024
          ReLU-21          [-1, 512, 4, 4]                 0
       Linear-22                  [-1, 10]             5,130
================================================================
Total params: 4,507,722
Params size MB: 17.20
----------------------------------------------------------------
```

```
ResNet-10:
----------------------------------------------------------------
       Layer type            Output Shape            Param #
================================================================
         Conv2d-1         [-1, 64, 32, 32]             1,728
    BatchNorm2d-2         [-1, 64, 32, 32]               128
         Conv2d-3         [-1, 64, 32, 32]            36,864
    BatchNorm2d-4         [-1, 64, 32, 32]               128
         Conv2d-5         [-1, 64, 32, 32]            36,864
    BatchNorm2d-6         [-1, 64, 32, 32]               128
     BasicBlock-7         [-1, 64, 32, 32]                 0
         Conv2d-8        [-1, 128, 16, 16]            73,728
    BatchNorm2d-9        [-1, 128, 16, 16]               256
        Conv2d-10        [-1, 128, 16, 16]           147,456
   BatchNorm2d-11        [-1, 128, 16, 16]               256
        Conv2d-12        [-1, 128, 16, 16]             8,192
   BatchNorm2d-13        [-1, 128, 16, 16]               256
    BasicBlock-14        [-1, 128, 16, 16]                 0
        Conv2d-15          [-1, 256, 8, 8]           294,912
   BatchNorm2d-16          [-1, 256, 8, 8]               512
        Conv2d-17          [-1, 256, 8, 8]           589,824
   BatchNorm2d-18          [-1, 256, 8, 8]               512
        Conv2d-19          [-1, 256, 8, 8]            32,768
   BatchNorm2d-20          [-1, 256, 8, 8]               512
    BasicBlock-21          [-1, 256, 8, 8]                 0
        Conv2d-22          [-1, 512, 4, 4]         1,179,648
   BatchNorm2d-23          [-1, 512, 4, 4]             1,024
```

```
        Conv2d-24            [-1, 512, 4, 4]         2,359,296
    BatchNorm2d-25           [-1, 512, 4, 4]             1,024
        Conv2d-26            [-1, 512, 4, 4]           131,072
    BatchNorm2d-27           [-1, 512, 4, 4]             1,024
     BasicBlock-28           [-1, 512, 4, 4]                 0
        Linear-29                  [-1, 10]             5,130
================================================================
Total params: 4,903,242
Params size MB: 18.70
----------------------------------------------------------------
```

LeNet style architecture for MNIST:
```
----------------------------------------------------------------
        Layer type            Output Shape         Param #
================================================================
        Conv2d-1          [-1, 20, 24, 24]             520
        MaxPool2d-2
        Conv2d-3            [-1, 50, 8, 8]          25,050
        MaxPool2d-4
        Linear-5                 [-1, 500]         400,500
        ReLU-6
        Linear-7                  [-1, 10]           5,010
================================================================
Total params: 431,080
Trainable params: 431,080
Non-trainable params: 0
----------------------------------------------------------------
Input size MB: 0.00
Forward/backward pass size MB: 0.12
Params size MB: 1.64
Estimated Total Size MB: 1.76
----------------------------------------------------------------
```

# G  ADDITIONAL EXPERIMENTS AND FIGURES

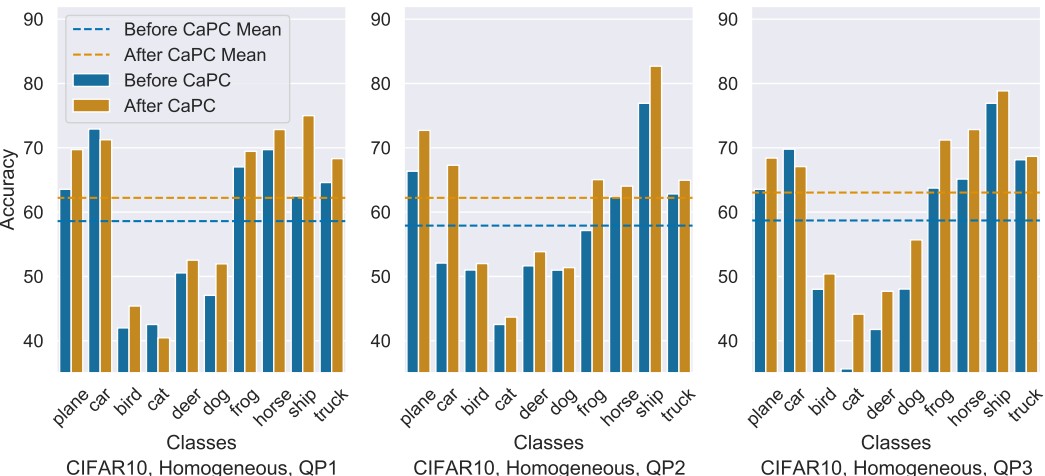

Figure 5: **Using CaPC to improve each party's model performance on the CIFAR10 dataset.** We observe that each separate querying party (QP) sees a per-class and overall accuracy bonus using CaPC.

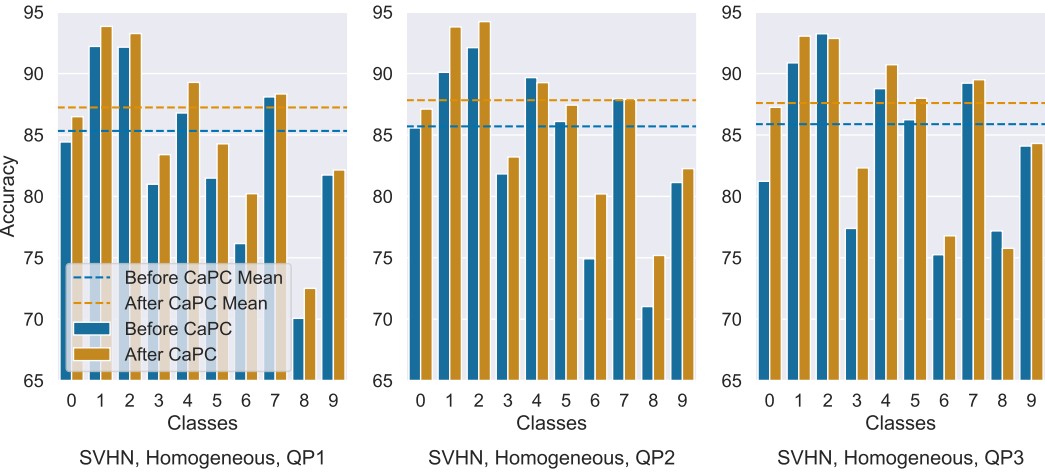

Figure 6: **Using CaPC to improve each party's model performance on the SVHN dataset.** We observe that all querying parties (QPs) see a net increase overall, with nearly every class seeing improved performance.

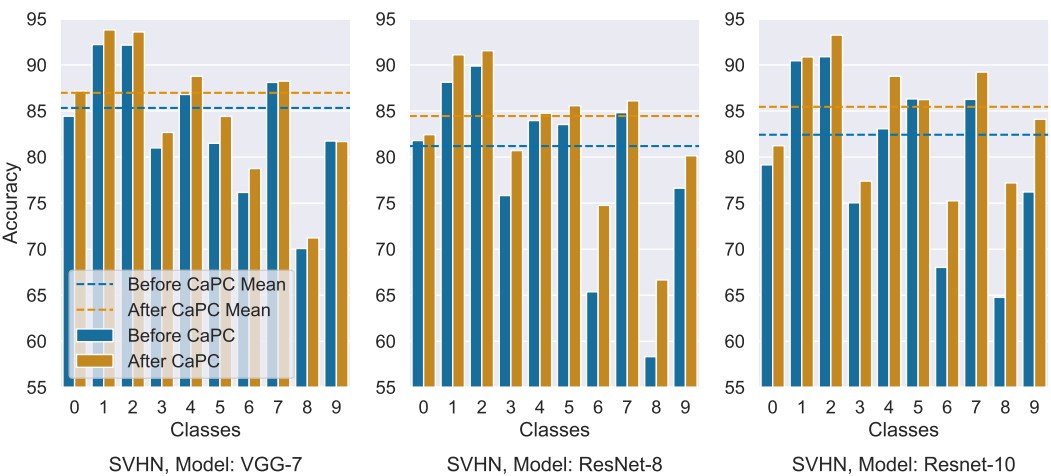

Figure 7: **Using CaPC to improve each party's heterogeneous model performance on the SVHN dataset.** Each querying party adopts a different model architecture (1 of 3) and $\frac{1}{3}$ of all answering parties adopt each model architecture. All model architectures see benefits from using CaPC.

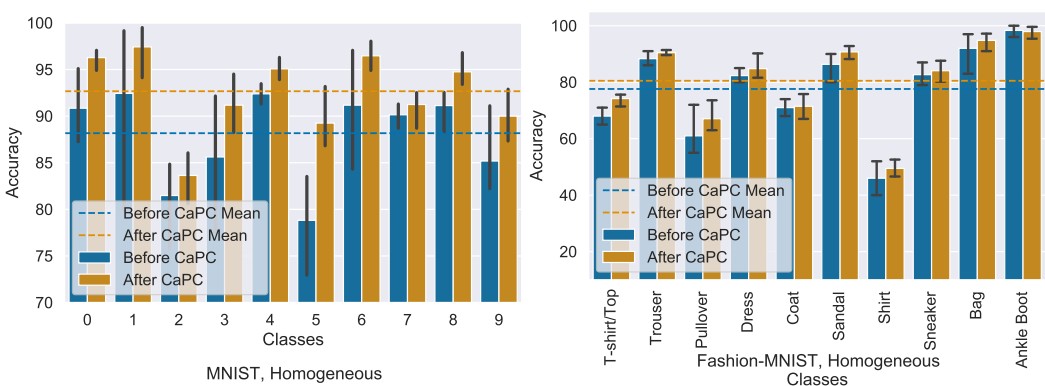

Figure 8: **Using CaPC to improve model performance on balanced MNIST on Fashion-MNIST.** Dashed lines represent mean accuracy. We observe a mean increase of 4.5% for MNIST ($\epsilon = 2.35$) and 2.9% for Fashion-MNIST ($\epsilon = 3.89$).

| Method | Forward Pass (Step 1a) |
|---|---|
| $CPU, P = 8192$ | $14.22 \pm 0.11$ |
| $CPU, P = 16384$ | $29.46 \pm 2.34$ |
| $CPU, P = 32768$ | $57.26 \pm 0.39$ |
| GPU, no encryption | $3.15 \pm 0.22$ |
| CPU, no encryption | $0.152 \pm 0.0082$ |

| QP-AP (Steps 1b and 1c) | QP-PG (Steps 2 and 3) |
|---|---|
| $0.12 \pm 0.0058$ | $0.030 \pm 0.0045$ |

Table 1: **Wall-clock time (sec) of various encryption methods with a batch size of 1.** We vary the modulus range, $P$, which denotes the max value of a given plain text number. Note that the GPU is slower than the CPU because of the mini-batch with a single data item and data transfer overhead to and from the GPU. We use the CryptoNet-ReLU model provided by HE-transformer (Boemer, 2020).

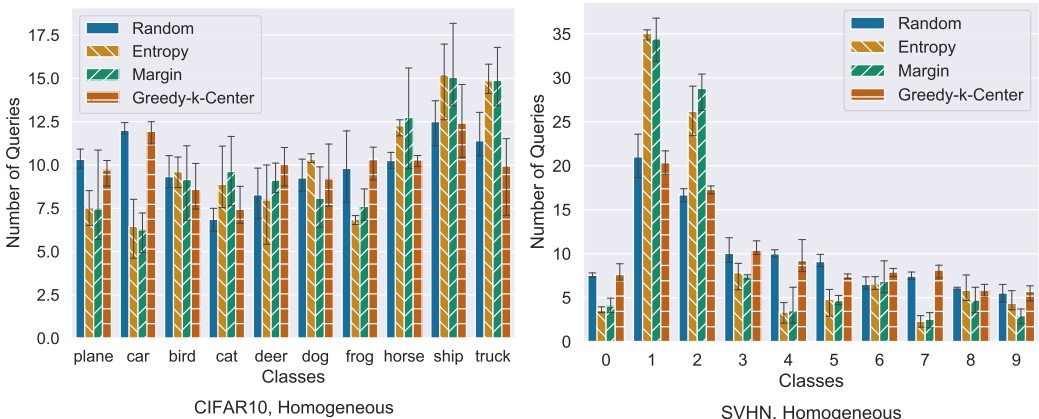

Figure 9: **Using active learning to improve CaPC fairness**. We observe that underrepresented classes are sampled more frequently than in a random strategy.

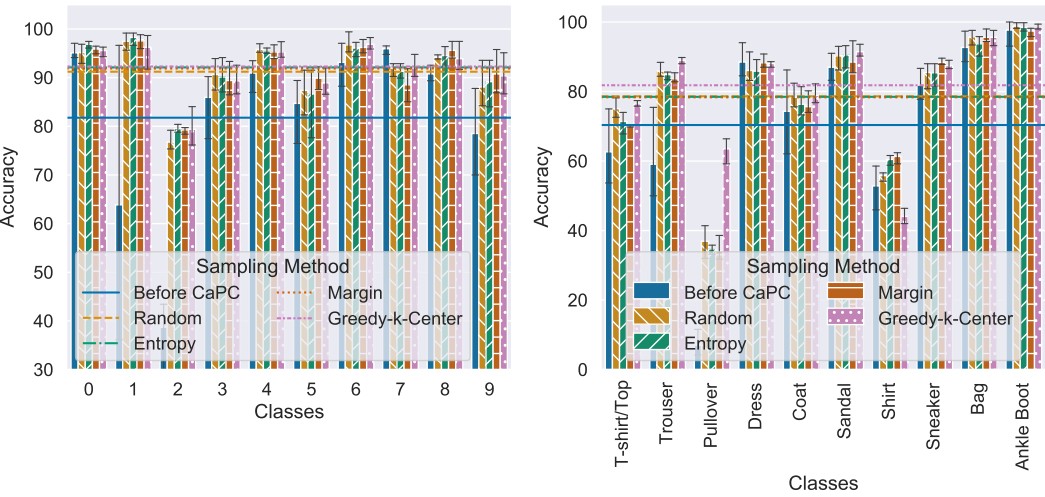

Figure 10: **Using CaPC with active learning to improve balanced accuracy under non-uniform data distribution.** *Dashed lines are balanced accuracy (BA).* We observe that all sampling strategies significantly improve BA and the best active learning scheme can improve BA by a total of 10.10 percentage-points (an additional 0.45 percentage points over Random sampling) on MNIST (left) and a total of 10.94 percentage-points (an additional 2.48) on Fashion-MNIST (right).

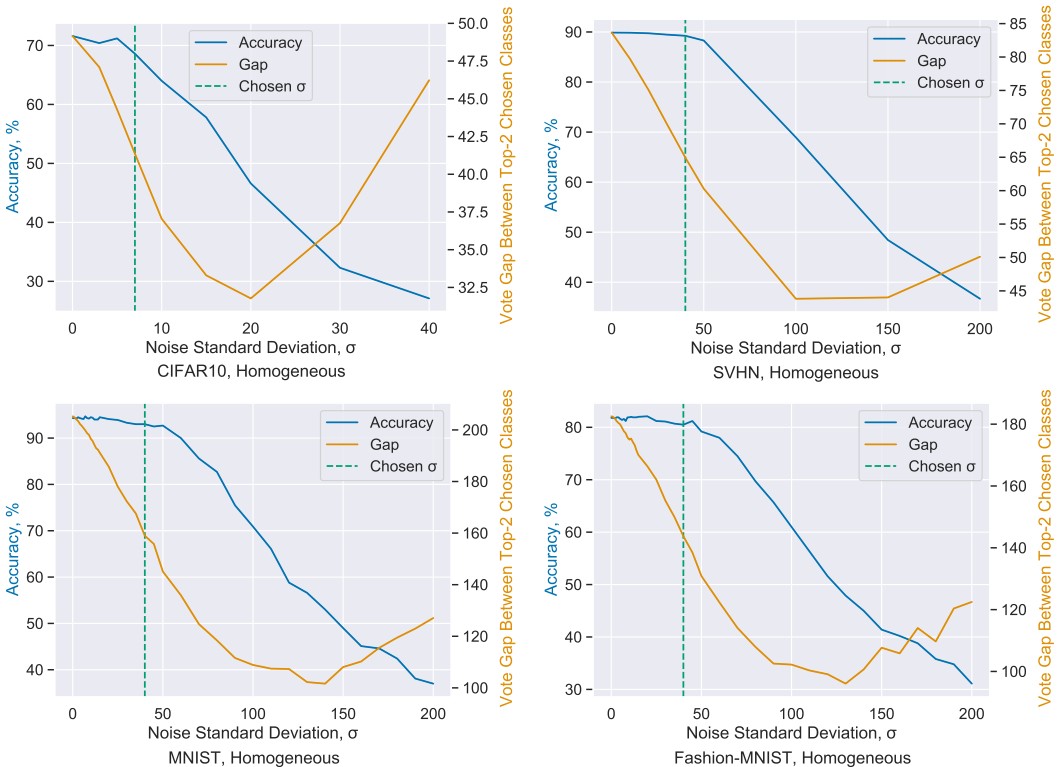

Figure 11: **Tuning the amount of noise ($\sigma$) in CaPC.** We tune the amount of Gaussian noise that should be injected in the noisy argmax mechanism by varying the standard deviation. We choose the highest noise: $\sigma = 7$ for CIFAR10, $\sigma = 40$ for SVHN, MNIST, and Fashion-MNIST, without having a significant impact on the model accuracy, allowing a minimal privacy budget expenditure while maximizing utility. We train 50 models for CIFAR10 and 250 models for SVHN, MNIST, and Fashion-MNIST.

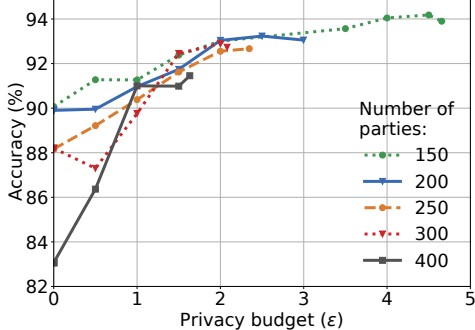

| | **Number of parties** | | | | |
|---|---|---|---|---|---|
| | 150 | 200 | 250 | 300 | 400 |
| **Accuracy gain (%)** | 4.11 | 3.33 | 4.50 | 4.69 | 8.39 |
| **Best $\varepsilon$** | 4.50 | 2.50 | 2.35 | 2.00 | 1.63 |

Figure 12: **Accuracy gain for balanced MNIST using CaPC versus number of parties and privacy budget, $\varepsilon$.** With more parties, we can achieve a higher accuracy gain at a smaller bound on $\varepsilon$.

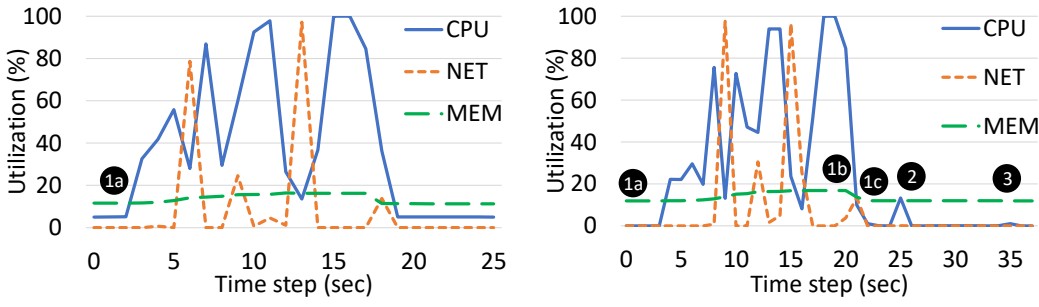

MP2ML (HE-transformer for nGraph).      CaPC (built on top of the MP2ML framework).

Figure 13: **Measuring the CPU, Network (NET), and Memory (MEM) usage over time for CaPC.** We use the CryptoNet-ReLU model provided by HE-transformer (Boemer, 2020) and sar (Godard, 2020) (System Activity Report) to perform this micro-analysis. We label the steps according to the CaPC protocol shown in Figure 1. The network usage reaches its peaks during execution of ReLU and then MaxPool, where the intermediate feature maps have to be exchanged between the querying and answering parties for the computation via garbled circuits.

