# OpenReview forum: "CaPC Learning: Confidential and Private Collaborative Learning"
_ICLR.cc/2021/Conference — ICLR 2021 Poster_

### Official Review · AnonReviewer4 · 2020-10-19
**CaPC Review. Decision: Accept.**

**Rating:** 7
**Confidence:** 4

**Review:**

This paper works on the problem of collaborative learning while preserving both confidentiality and privacy of the data points. It combines techniques from secure multi-party computation and differential privacy for the same, and improves on confidential inference and PATE in the process. The new technique is called CaPC. Finally, it states empirical results as evidence for the improved accuracy.

Weakness:
1. The evaluation is done on just two datasets. So, it is a little hard to judge whether the techniques would generalise or not.
2. The writing of the paper itself is not that great because it is difficult to understand the low level details of the experiments.
3. They talk very little about improving on the fairness guarantees.

Strengths:
1. Their techniques enable collaborative learning even in settings where the local architectures of different parties are different.
2. The algorithms they provide improve on fairness.
3. Their empirical results are better than the previously known methods.

Evaluation: I believe the combination of secure multi-party computation and differential privacy is not totally new, but since it yields decent results, I would say that the paper deserves a chance to be accepted.

---

> ### Author Response · Authors · 2020-11-13
> **Added results for the MNIST dataset and more information on experimental details as well as on fairness**
>
> Thank you for your careful analysis of our paper.
>
> Regarding weakness #1: To confirm that our approach generalises, we added results for the MNIST dataset (please see Section F.1 and Figures 8, 10, and 12).
>
> Regarding weakness #2: We elaborated on the low level details of the experiments in Appendices F and G. We added more information about the setup in the new version of the paper in Section 4.2. If there is something else that you would like us to add, let us know and we will be happy to include additional details. We would also like to indicate that our source code was released along with our initial submission to facilitate reproducibility of our results.
>
> Regarding weakness #3: We also expanded our discussion on fairness, including more background on techniques for improving fairness and more description on how these can be incorporated into CaPC (see Sections 4.3.2 and 5).
>
> Regarding the overall evaluation: though we agree that neither multi-party computation and differential privacy are new, our work is the first to combine the two to ensure both confidentiality (of the test inputs and model parameters) and privacy (of the training data) in a collaborative learning setting where (a) the parties each trained heterogeneous architectures and (b) there are few parties participating in the protocol.

---

### Official Review · AnonReviewer1 · 2020-10-19
**Fairness seems very cool, but no convinced by privacy part**

**Rating:** 7
**Confidence:** 4

**Review:**



This work motivated by healthcare and finance where separate parties may wish to collaborate and learn from each other's data but are prevented from doing so due to privacy regulations. This paper propose Confidential and Private Collaborative (CaPC) learning, the first method provably achieving both confidentially and privacy in a collaborative setting. This work also discussed about fairness. I liked this part, since it seems very cool. However, I'm not convinced by the method in this work is better than InstaHide (I could be wrong).




Minor comments
In Section 2.1, this paper should be discussed, since it proposed a way to ``encrypt'' the images/texts.

InstaHide: instance-hiding schemes for private distributed learning
https://arxiv.org/abs/2010.02772
ICML 2020
Yangsibo Huang, Zhao Song, Kai Li, Sanjeev Arora.


TextHide: Tackling Data Privacy in Language Understanding Tasks
https://arxiv.org/abs/2010.06053
EMNLP 2020
Yangsibo Huang, Zhao Song, Danqi Chen, Kai Li, Sanjeev Arora

In Section B, it lists many theorems/definition about differential privacy. In Section C, it list many backgrounds about sampling. In Section D., it list many definitions on Fairness. I don't quite see the point of having them in appendix, since none of them got mentioned in Appendix E, which is the proof of the main theory result in this paper.

This paper is closely related differential privacy. I think this paper should also be mentioned somewhere.

Privacy-preserving Learning via Deep Net Pruning
https://arxiv.org/abs/2003.01876
Yangsibo Huang, Yushan Su, Sachin Ravi, Zhao Song, Sanjeev Arora, Kai Li.

---

> ### Author Response · Authors · 2020-11-19
> **Related work & appendices**
>
> Thank you for your feedback, we uploaded a new PDF that discusses the papers mentioned in your review.
>
> We find the InstaHide work interesting. We now clarify in our paper that we achieve confidentiality at test time. Instead, InstaHide modifies the model during training and is thus orthogonal to our approach. The two could even be combined. In addition to the guarantees of confidentiality provided by the use of cryptographic primitives, our protocol also provides guarantees of differential privacy through the noisy argmax mechanism. Confidentiality and differential privacy are orthogonal and complement one another.
>
> Thank you for bringing the paper [1] to our attention. We now outline in our revised manuscript the difference between our work and [1]. In our work, differential privacy guarantees are obtained by adding noise to the histogram of aggregated one-hot-encoding vectors predicted by each answering party (i.e., the noisy argmax mechanism) whereas the approach in [1] obtains these guarantees by pruning an internal layer of a neural network. This means that new analysis is required for each layer type being pruned. For instance, Theorem 3.2 in [1] is stated for a fully-connected neural network. Instead, our approach is applicable to any architecture (e.g., our experiments also use fully convolutional neural networks) and furthermore no changes need to be made to the way the architecture is trained.
>
> Regarding appendices B, C, and D, we wanted to make sure that the paper is understandable to a wide audience of people, e.g., those with only a machine learning background. Thank you for your close attention to their usage. In our updated manuscript, we have added additional pointers to these appendices to make better use of them and clearly delineate them from Appendix E. The following are our updates:
>
> a) Although appendices C and D do not pertain to our result in Appendix E, they provide background on active learning and fairness that we use in our experimental evaluation (Sections 4.2 and 4.3.2, and Appendix G). We have added forward pointers to Appendix C and D in section 4.3.2 which comprises the bulk of our exposition into fairness and active learning within the CaPC setting. We also use these definitions in our updated discussion (see Section 5).
>
> b) We have added forward pointers to Appendix B in Sections 2.2, 3.3, and 4.4, referring readers for more details on differential privacy, PATE, and the calculation of epsilon, respectively.
>
> [1] Yangsibo Huang, Yushan Su, Sachin Ravi, Zhao Song, Sanjeev Arora, Kai Li. "Privacy-preserving Learning via Deep Net Pruning" https://arxiv.org/abs/2003.01876

---

### Official Review · AnonReviewer2 · 2020-10-27
**Solid but unsurprising system for federated classification system with privacy**

**Rating:** 7
**Confidence:** 4

**Review:**

Summary:
The authors combine several cryptographic techniques to create a federated systems that allows several entities to run classification against all the model held be the participants without revealing information in the process. In particular, the sample to be classified is not revealed to any other party, and differential privacy is used to protect the training data that was used to train the models. A central semi-honest coordinator is used to aggregate the results and add the differential privacy without learning any private information.

Pros:
The strength of this works lie in combining relevant techniques and to show experimentally that the resulting system does improve over using a local model both when the training is distributed evenly or in skewed manner while taking privacy considerations into account.

Cons:
- From a cryptographic point of view, the combination of techniques is somewhat expectable.
- I'm wondering about the low statistical security (${2^-23}$). This seems to be related to the usage of (unbounded) integer secret sharing. Would it be possible to use secret sharing modulo an integer, in which case the security could be perfect?
- I think it would be easier to follow if steps 1-3 were combined in the description because they all take between the same pairs of parties. The exact techniques used don't seem to matter as long as the output secret sharing is the desired result, namely the one-hot vector.
- I find the term collaborative learning somewhat overblown because the proposed protocol only runs classification collaboratively.

Overall:
Despite the points above, I'm in favor of acceptance because the paper seems to improve on previous work, and because it is written very well.

Minor issues:
3.3: leakeage
4.1: odd juxtaposition in the formatting of "arg max" and "sum"
Figure 3: very hard to read in black-and-white

---

> ### Author Response · Authors · 2020-11-13
> **CaPC protocol & statistical security**
>
> We thank the reviewer for the insightful feedback.
>
> Regarding statistical security, our protocol design supports high statistical security. The low statistical security in our current implementation is a consequence of the HE-transformer not supporting label predictions that forces us to operate over logits, which are unbounded as you indicate.
>
> The secret sharing step of our implementation can achieve perfect security by using properties of the underlying fully homomorphic encryption scheme. We have confirmed this with the authors of the HE-transformer and are working to integrate this into our implementation. We estimate that the fix will not incur any slowdown or error to our implementation.
>
> Indeed, steps 2 and 3 of the protocol could be incorporated inside a private inference library. To make our protocol flexible to any private inference library, not just those that return the label predicted by the model (some only return logits, e.g., the HE-transformer), we decided to incorporate these steps outside of the library and within our protocol. We clarified it in the updated version of the paper in Section 4.1 and will continue to improve the description of the protocol in Section 3.2.
>
> Regarding the term collaborative learning, we indicate in Section 3.1 that collaboration here refers to the different parties collaborating on classification. One of the main benefits of this collaboration is that it can be used by each party to train a local model (independently of its architecture) with improved performance.
>
> Thank you for carefully reading our paper. We corrected the typo in Section 3.3, applied the same formatting to “argmax” and “sum” in Section 4.1, and added patterns to columns to increase the readability of Figure 3. We have applied this styling to the other Figures similarly.

---

> > ### Author Response · Authors · 2020-11-17
> > **Perfect secret sharing and renumbered steps of the protocol**
> >
> > We have updated the implementation of CaPC to use secret sharing over a prime-order field to achieve perfect secret sharing. It was made possible by using underlying properties of the CKKS encryption scheme: although the ciphertext operations are rational, the underlying ciphertext space is still a field. Our fix only needs to use an appropriate secret sharing and does not involve any extra cryptographic step. As a result, it does not have any impact on the speed or the accuracy of our implementation.
> >
> > Following your suggestion that “it would be easier to follow if steps 1-3 were combined in the description”, we changed the protocol and the steps from 1 to 3 are now numbered as 1a, 1b, 1c, where we obtain the one-hot encoded results (in step 1). The remaining (previous) steps 4 and 5 are now denoted as 2 and 3. Now, a reader can understand the overall protocol without delving into the details of steps 1a, 1b, and 1c.

---

### Decision · Program_Chairs · 2021-01-07
**Final Decision**

**Decision:**

Accept (Poster)

**Comment:**

This work describes a system for collaborative learning in which several agents holding data want to improve their models by asking other agents to label their points. The system preserves confidentiality of queries using MPC and also throws in differentially private aggregation of labels (taken from the PATE framework). It provides expriments showing computational feasibility of the system. The techniques use active learning to improve the models.

Overall the ingredients are fairly standard but are put together in a new (to the best of my , admittedly limited, knowledge of this area). This seems like a solid attempt to explore approaches for learning in a federated setting with strong limitations on data sharing.